# Semantic Digital Twins for Organizational Development

Alfonso Díez[1] and Juan de Lara[2]

[1] UGROUND GLOBAL, Madrid, Spain
[2] Universidad Autónoma de Madrid, Madrid, Spain

**Abstract.** Semantic-based Digital Twins can be applied to the enactment of the digital counterparts of enterprises and organizations, and not only for physical devices or machines. Digital Twins for Organizations have a profound impact in the architecture of corporate management platforms, in the design of business processes and in the conception of information and technological ecosystems.. This work condenses our experience in the creation of Digital Twins for Organizations, using model-driven engineering based on semantic modeling. We will describe their more relevant characterizations and discuss their potential impact in their business implementations.

**Keywords:** Digital Twins for Organizations, Digital Twins, Model Driven Engineering, Semantic Engineering, Semantic Digital Twins, Digital Transformation, Digitalization, Enactive Systems

## 1    Introduction

The idea of Digital Twins (DTs) [7,10,11] belongs to the world of Digital Transformation. This is the transformation of the economy and society to a new paradigm, where digital means are pervasive.

This paper distils our experience at UGROUND GLOBAL[1] with the concept of DTs for organizations (DTOs), characterizing their essential elements, and discussing their potential impact. We started our theoretical work and implementation of solutions for digitalization from the point of view of Model-driven Engineering (MDE) [3], using semantic approaches and interpretation of models. Soon we found a more complex need: large organizations require complex and evolutive solutions to cover many domains of their operations, which is not possible with the conventional approach of niche software applications that has dominated the market. We realized that we were searching for DTs of entire enterprises, and that this idea could not be fulfilled if we think that digitalization of organizations is a matter of piling more and more conventional applications. Digitalization, digital transformation and DTOs are interlaced concepts that suggest a new challenge: the development of universal platforms configured using semantic modelling to create DTs of arbitrary large organizations. These platforms are not *yet another application*, but a class of meta-

---

[1] https://www.uground.com/

applications that can represent many different views of reality and evolve very fast, interacting with real objects, people and other technologies. Such interaction between the DT and the reality it represents is dialectic, each one changing the other, in search of an impossible equilibrium. These universal and non-deterministic views are currently missing in current views of the DT as the digital simulation of the *real* organization [10]. In the rest of this paper our aim is introducing such novel view of DTOs.

## 2    A systemic approach to Digital Twins

A DT is, in its common definition, the virtual replica of an asset or physical entity [7,10,11]. However, this definition is today being extended to other new areas: from digital representations of physical objects to digital representations of whole organizations [10] or digital replications of living and nonliving entities [11].

This virtual emulation done by the DT serves different purposes. The primary one is to avoid the need for creating the physical entity to test its behavior, safety, or usability before it is built. However, when we have the digital model of a physical entity, we can interact with such an entity in a different way, not only enriching its behavior but changing its nature. One topical example are modern cars. Thirty years ago, cars were mechanical entities, managed by the driver. The driver took care of all aspects of the car, such as its security (opening and closing doors, avoiding crashes), the efficiency in driving, the mechanical maintenance, the diagnostic of mechanical conditions, the provision of supplies like gas or tires, insurance and location. Nowadays the situation is very different; cars are increasingly "*computers with tires*" that incorporate functions previously reserved for humans. A modern car takes care of its physical security, not only when stopped, but when driving and parking, and the security of passengers; the car manages its location, its own mechanical diagnostics, and its maintenance needs, even scheduling appointments with the physical mechanical shop. It is also able to contract insurances, to report accidents. The trend, in a foreseeable future, is that cars will drive themselves, and the driver will be the ultimate user that gets the final value: being physically transported from one place to another [13].

The conclusion of the preceding description is that the car has incorporated a DT of the system car-driver. Now, the car is a triplet: the driver, the mechanical device and the logical system that interacts with and controls the behavior of the other two. It is important to point out the relevance of the continuous flow of information between the DT and its physical concretion.

Now, let's consider another example of a very different nature: an enterprise. We can say that an enterprise is a bundle of physical and logical entities managed by people. An enterprise has buildings, warehouses, products, rules, customers, money and bank relations, rules and procedures, trucks, organizational structures, information systems, computers, etc. As a physical structure that is operational we can consider, rightly, if the concept of DT could be applied to it. We will see, in the rest of this paper, the deepness and importance of such quest.

## 3      The nature of an enterprise

An enterprise is, above all, a social system; it is a human endeavor that creates its mission and obtain the resources for it. Every company in the world is, above all, an arbitrary mixture of interests, resources and knowledge that will drive its future [5]. Every existing company is tangible, even those running the most advanced digital platforms such as Google or Facebook: they are people-based and have a physical existence. A company is not a dumb machine; it is not a black box with a number of blind rules to follow, like a car. A company is a highly volatile system that is interacting with the environment and changing at high speed. This creates unique challenges for creating DTs; on one hand there is a physical device to emulate, but on the other we have to ask ourselves which rules the DT needs to acquire.

To be able to advance in this question we have to ask, more deeply, about the precise nature of an enterprise. We can define an enterprise as a social generative system whose functioning is based on financial restrictions. It does not matter if it is a small shop or a multinational, if it is for-profit or not, or if it is public or private, it will always be defined by its social fundaments (driven by people), its self-generative characteristics (the enterprise develops itself, looking internally or externally for the development resources, whether knowledge or money), and by the fact that it is constricted an ruled by the availability of money (capital, cash, loans, profits, etc.).

The basic foundation of an enterprise is based on two components: The Social System and the Physical Setup, as it is shown in Fig. 1.

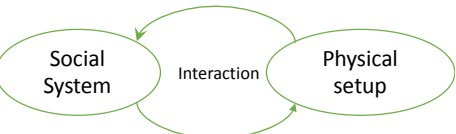

The Social System and the Physical Setup are the conventional components of the enterprise that support each other

*Fig. 1. Basic infrastructure of an enterprise: The Fundamental System*

in a symbiotic way. We will call them the Fundamental System of the company (FS). The Physical Setup includes not only buildings or factories, but also computers, programs, data, rules and procedures, relations, customers and suppliers, financial resources, patents and intellectual property, among others. This physical setup is the supporting set of resources for the enterprise. The other component is the Social System that has, as its main component, the People that belongs to the enterprise, work for it or have strong relations with it. Here we are counting not only the persons, but also their abilities and effort at work, their talent, knowledge, creativity, relations, ideals and moral attitudes.

Both entities, Social System and the Physical Setup, develop themselves in a recursive way. They will try to enrich, upgrade and grow as part of its life cycle. The Social System will grow not only adding new people, but also training the team, creating new teams and organizational structures, improving ideas and innovating, researching in the quest for new knowledge, etc. The Physical setup will improve or increase its components. In this cycle, products and labor will be transformed into financial resources, that are, in turn, transformed into factories, buildings, and other resources.

There is a third component in organizations, which is embedded in the other two in terms of resources but separated in its purpose: the Governance System (GS), shown

4

in Fig. 2. This system is in charge of structuring how the FS works. The usual components of this system are the Board and the executive organs, the hierarchical-organizational structures, the organizational rules and procedures, the different quality assurance mechanisms (ISO, FQM, and so), and the reporting, planning and strategical facilities and rules.

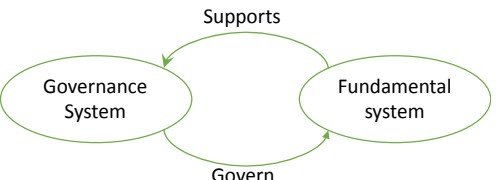

*Fig. 2. Governance System coupled with Fundamental System*

The Sciences of Management are devoted to the creation of this GS, and it is well known that is a difficult task that is key for the success or failure or the enterprise, perhaps more than any other. Every company dreams of having a fluid control, a good management structure and team, and firm and nimble strategy, but few actually have it.

An enterprise is a supercritical system because it self-generates its own behaviors and missions. It is not a trivial machine or a standing-still organism (if it would be so, then it would be done, because it will not adapt to the evolution of the ecosystem).

Because of its inner nature, an enterprise has specific features that will not be found in other systems. Let's review these in detail.

The enterprise is a *semantic system*; it is described, defined and characterized using expressions that are meaningful for the humans that are part of it. Every enterprise has an internal lingua that define the concepts, adjectives or verbs that are used within the edge of the enterprise. Every enterprise can have a *semantic* representation of itself, where the conceptualization of the structure and its functions are well defined. This representation is done by humans analyzing its functioning and mission. The transformation of the enterprise into a semantic system is possible because everything in the enterprise can be described in terms of propositions.

For instance, when we say that "*the accounting clerk checks the payments against pending invoices, marking them as paid; when an invoice reaches its due date without payment it becomes unpaid*" we are describing, in a function-structure analysis, different meta-models such as organizational elements (department and role), entities (payment and invoice), states (invoice pending, paid and unpaid) and events (payment received, due date reached). In summary, every company can be mapped into a set of propositions and, therefore, into a formal model.

Compared with other systems, the enterprise has unlimited degrees of freedom from the organizational point of view. This is because its capabilities to generate emergent behaviors, that allows it to experiment new organizational rules and structures with no limit. There is no possibility to fix the internal knowledge of the company about itself in a fixed moment of time. This is caused by the application of the Heisenberg principle: the interaction of the analyst that is documenting the behavior of the company with the analyzed object changes its behavior, as consultants experience in their daily work.

A company is, therefore, a highly dynamic object; its behavior varying continuously depending on internal or external motivations, like markets, regulations, logistics, and consumer attitudes. There is no way to establish rules of certainty about the future

of a company. The very existence of an enterprise depends on people, which are the agents of the company's present and future, and the interpreters of its past. This means that is volatile by nature, because people's opinions cannot be foreseen.

In summary, we have characterized an enterprise as a volatile, unpredictable system that can be explained using semantic descriptions. There is no way to create a fixed, full description of an enterprise, and the number of degrees of freedom suggest that the representation of the company has to be generative.

## 4     Digital Twins of Organizations represent social structures

We have described a conventional enterprise as a dynamic combination of a FS (a Social System and a Physical Setup) and a GS. If we want to transform this organization into a digital one, we need to add the digital view of the company as a generalized resource platform that will acquire the data and rules of the enterprise. This is the Digital Twin of an Organization (DTO). Fig. 3 depicts schematically the structure of the enterprise and the relation with the DTO. The self-loops indicate that each component generates its own new capabilities and configurations along time.

The DTO is the third entity of the picture. It is the virtual counterpart of the conventional enterprise (FS-GS) and is able to emulate its behavior as a system. This means that the DTO is not one more application in the physical setup, but a platform with its own existence and autonomy. The DTO is in charge of analyzing and controlling the conventional enterprise to improve its performance in all terms: from the ability to hire employees and resources to the planning of production and the usage of materials, or the financial schedule and

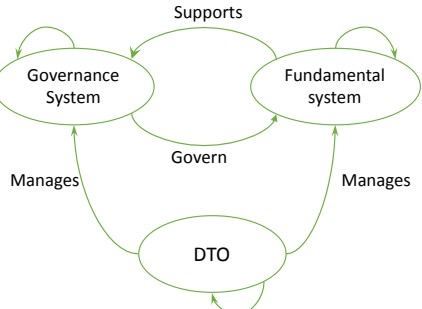

*Fig. 3. The DTO manages both the FS and the GS. Self-loops indicate that each component self-generates its own evolution.*

control of performance indicators. The DTO will interact continuously with the enterprise's resources and their dynamics, whether human resources, physical, technical, and governance.

Of course, in the conventional enterprise we find many software packages that perform data management and process control. They are technical resources that play the role of functional black boxes for some well-defined tasks. These conventional software packages are the silo functional applications and standard utilities such as office tools or mail, which are the common logical support for the company's activities for sending mails, generating billing, storing documents, or producing payroll. It seems that there could be a relevant overlapping between this software and the DTO. The DTO approach will not replace these conventional tools but will interact with them to extract information or manage events, in the light of the functioning of the enterprise system. That is, DTO orchestrates every resource, even if it is a software tool.

From a cybernetic point of view, the DTO is a control system that uses positive and negative feedback to improve the functioning of the conventional enterprise system. For this, the DTO uses mechanisms of sensors and effectors in the enterprise. On the human side of the enterprise (social and governance layers), the interface is resolved using (usually) desktop and mobile applications, to push information and tasks to humans and to get information from them. In the Physical Setup, it will use two different types of components. The first one contains physical sensors and actuators, using IoT technologies. The other are technical interfaces between systems, which use technologies such as SOAP, REST, APIs, protocols, files, data bases, sockets, message queue, RPA, and others.

The DTO is the control system of the organization's operations. Therefore, the DTO has to implement the capability to control itself, as it is one of the key control resources of the organization. Next we review its main features.

1. **Semantic-based technology.** A DTO is a technology able to interpret the specific semantic definition of an enterprise. It is important to remember that, because the enterprise is not a machine but a supercritical system, the only possible description of an enterprise is generative semantics, not algorithmic, deterministic models. In other words, algorithm-based conventional technologies cannot create DTOs. The capability to interpret the human-generated semantic view of the enterprise means that the DT emulates the enterprise, up to the level of precision of the description, and therefore it behaves as if it were the digital counterpart of the physical one.

2. **Self-referential – Reflective.** Since the DTO is a part of the DTO definition itself, it needs to have an explicit representation. This means that the DTO has to interact with itself not only as an executing entity, but also in terms of its own rules and concepts. This enforces the description of the DTO as a generative semantic device able to interpret entities and rules of the outer world, and its own semantic description. This has a relevant consequence whereby the interpretation engine used by the DTO for its application is the same as the one for its own internal specifications. In other words, there is only one technical engine to interpret semantic descriptions, whether we are building a DTO or end-user applications with it. Hence, the engineering environment that creates the DTO infrastructures, the development environment that creates end-user applications, and the production environment that executes those applications are three instances of the same technical artifact. This has an additional consequence: the separation of the implementation engines (interpreters) from the semantic definitions (the knowledge), so that the implementation engine can address a subset of definitions that can be its own internal description, its own performance, or its application to external problems [6].

3. **Recursion.** The self-referential property of DTOs implies that any DTO is recursive (generative) and is able to create new meanings reusing the existing ones. The recursiveness and the semantic design suggest that workable descriptions of DTOs can be based on ontologies [14], and that DTOs are semantic representations of human knowledge. A DTO has a kernel of concepts (the bricks of the ontology) that is the minimum set of concepts that generate, in a recursive manner, the DTO itself and its applications. The kernel has a technical implementation that allows

concepts to interact with the outer world (FS-GS). A concept can be a functional term, such as *invoice*, or can have a technical description and implementation, such as *email, blockchain* or *geometry*. The DTO is elastic and able to allow the addition of new concepts, and perhaps its technical implementations, at the same time that FS-GS changes, and to provoke changes in FS-GS systems. Therefore, DTO is a technology that has generative properties [1]. Clearly, the representation of the knowledge in DTO covers both the functional and technical worlds, which has implications in the profiles of the engineers that will evolve this knowledge base.

4. **Enactive design**. In the field of Model-driven Engineering (MDE), models are the description of the system to be enacted [3]. Enaction is the mechanism that brings into real existence something that was only a possibility [15]; in our case, a model comes into real existence driven by an interpretative technology. In most current MDE technologies, enaction is usually done through code generation. In this case an application is generated using the model as the set of specifications. Because of the ontological nature of the enterprises, the models that govern a DTO have to be declarative and rule-based, rather that fixed algorithms, and therefore the enaction has to be done using real-time interpretation of the ontological models.

The DTO has the capability to implement ontological descriptions of the enterprise reality. This reality is composed of functional aspects (enterprises business rules) and technical aspects (technical implementation and resources). Therefore, the DTO needs the capability to describe, using ontological methods, the functional and technical models of the company. The formal description of models, editors, monitors, and model interpreters can be done using the own modeling resources of the DTO. The consequence is that the DTO has a representation of the outer world in terms of models, and also another representation of its inner structure as recursive ontologies. The knowledge representations that form the models are edited using knowledge editors, which are ontological models to represent abstract archetypes [2]. The interpreters of these models are, in turn, other ontological models that are able to understand and enact the definitions previously expressed. Any ontological concept can be an enacted instance or an archetype or both.

In summary, the DTO is an enactive framework, a single entity based on knowledge representations and interpreters that execute rules and connect knowledge with the FS-GS using different types of human and technical interfaces.

5. **Self-organized.** As we have seen, a DTO is required to adapt itself to changing conditions [12]. The state of the DTO depends on, and interacts with, the state of the FS-GS system, and will evolve not in an algorithmic, deterministic way, but reacting to the dynamical evolution of FS-GS, the relation of FS-GS with DTO, and the influence of DTO to itself. Therefore, the DTO needs to have self-organizational features, triggering adaptations or requests for adaptations. This self-organization is based on the analysis of its own performance in three senses:

- measuring its own *activity* using logging and monitoring: do I have disk space? how many users do I have? is my response time acceptable? etc.
- measuring its own *complexity* it terms of its own models and the applications models developed for the enterprise using the DTO.

- measuring the *performance indicators* of the enterprise in terms of sales, production, supply, treasury and all other parameters that are key for the FS to success. The performance indicators are multidimensional time series that describe any indicator in terms of time (each day, each end-of-month, etc), and dimensions such as branch, product, vendor, customer type, etc.

Both measures can be analyzed in terms of correlations, forecasting, anomaly detection, and other methods of advanced data analytics. The results of these analyses become a source of information for humans, and for the DTO to decide its own internal states.

Self-organization does not mean that a DTO is able to generate and implement its own behavior. This capability is related to the ability to generate new knowledge, which is outside of the state-of-the-art and of the scope of this research. In fact, DTOs can be provided with ontologies that implement self-describing concepts, such as, for example, *Profile of User Activity*, that can adapt the performances of the DTO in a flexible manner, or can generate information for the *knowledge engineers* that maintain the representation of the DTO world.

6. **Autonomous.** The final conclusion of the already described properties is that the DTO is an autonomous representation of the DTO-FS-GS system. The DTO is able to self-manage aspects such as performance, scheduling, diagnose, or deployment. However, this autonomy has limits, because of the ontological nature of its description. An ontology is a way to structure knowledge, and knowledge is, until today, a human capacity not found in machines. No machine, nor the DTO, can create new concepts or archetypes, or decide its details. This is a capability reserved to humans. Therefore, the DTO will pair with humans to extend or change its functioning, using its self-referential design to manage its own requests for change, whether requested by the DTO itself or by humans.

In summary, a DTO is a class of technology by itself; it is an *enactive* platform that has a *self-referential* semantic kernel that grows recursively to address universal complex problems, producing a *self-organized* and *autonomous* behavior. In this way, a DTO is a propositional system coupled with an enactment engine, that is, in some way, a prototype of the mind-language system. DTOs have the properties of identity, continuity, structural adaptation and operational closure that make them autonomous systems [8,9]. In the same way, the DTO-FS-GS is a system-of-systems that interacts with the outer world to accomplish its mission. DTOs are fundamentally self-organized systems in many aspects, however they have to interact continuously with humans to create and evolve the enacted representations. The self-organization capabilities are today an area of research with large opportunities for this field [12].

## 5     The impact of DTOs

The DTO changes the way the enterprise sees itself. Therefore, a DTO means a profound change in perspective for enterprise managers. If we take a look at how a company manages its operations, we see that there is a continuous struggle for mech-

anizing activities, to reduce complexity and chaos, and improve order. Operations are always in constant tension in between the need for creating variants and innovating processes and products, and the need for reducing the noise that is generated regulating and industrializing such variants. This is frequently expressed by IT people when they complain that "*users never know what they want*". This is an exact expression, because the role of business users is to create variants of the procedures, and the role of IT is to reduce them to abstractions and predictable rules. The mechanistic view of the company simply does not work nowadays, where changes are so fast and wide.

For this reason, DTs of the conventional enterprises are crucial tools to implement new management rules, allowing companies to cope with the complexity of the social and economic world. DTOs, then, have a profound impact in enterprise operations and management, which we review next.

## 5.1   DTO and Digital Transformation

Digital Transformation is the process of taking a conventional enterprise (as an FS-GS pair) and transforming all its relevant processes (operational, support and management) to have a digitized support in its design, execution, control and management. Clearly, digital transformation is not the process of adding many new applications and digital devices to the enterprise, but to create a digital framework that takes digital control of all resources. In other words, digital transformation succeeds by creating a DT of the organization as shown in the next figure.

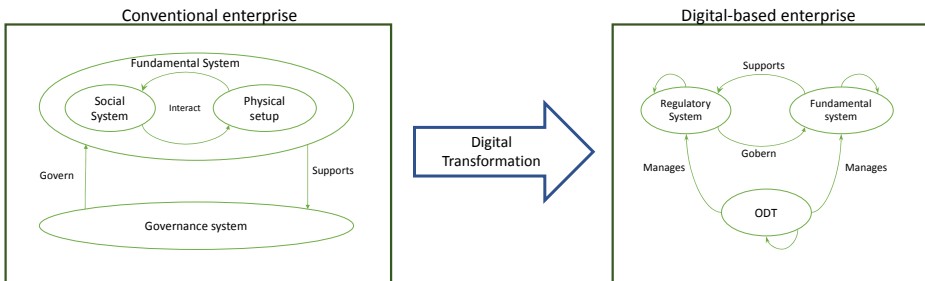

*Fig. 4 Digital transformation is based on the addition of a DTO that creates a virtual representation of the conventional enterprise, that is used to control its behavior in all of its features*

This conception of the DTO as the core mechanism in the process of digital transformation has important consequences in the way the company will be managed.

Firstly, DTO means to reposition the center of gravity of the organization. In a conventional enterprise the management is done by humans. They know (or should know) what things need to be done, why, and how. If we want to know something about the company, we have to ask humans, they will look into their data and documents, knowledge and records to produce the answer. They will control not only the information, but also the rhythm: they manage the clock that governs the dynamic part of the organization. Digital transformation means that the organization is going to give this responsibility to the DTO. The DTO will know (as the semantic repository) the concepts and the rules, will monitor the activity and set the clock of the processes

and its coordination. In summary, the DTO will become the center of gravity and will cooperate with the FS-GS to improve its functioning.

Here we reach the second part of the impact of digital transformation. The cooperation between the DTO and the FS-GS is of mutual dependence. Each time we improve the representation of the FS-GS into the DTO we are changing the way the DTO and therefore the FS-GS, which in turn will change the DTO, and so on until we reach a temporary stable state. However, the need for the organization to adapt to the changing outer world in a structural adaptation process (well described by the systems theory) means that these mutual changes will continue in the future with no end.

Here it is worthwhile to paraphrase the famous quotation from Lord Kelvin: "*What is not defined cannot be measured. What is not measured, cannot be improved. What is not improved, is always degraded*". The DTO is the way we define the structure and function of the organization, and therefore we can use a DTO to measure it. The act of measurement allows us to improve the DTO (incorporating innovations into it), avoiding the degradation that comes from the natural entropy of organizations. The DTO becomes the steering wheel of the organization, and the gauge device to control its situation in real time.

## 5.2    DTO and functional systems

The question of what is the relationship between the DTO (as a computer system) and the conventional information systems that we use today arises in a natural way, to solve functional tasks

Conventional technologies will never become DTOs because of the basic requirement that a DTO is a universal adaptive system, and therefore they are not deterministic, niche-oriented solutions. However, as conventional technologies implement algorithms for enterprises that are common and useful, they will keep being used in the future, which arises the question of the relationship between them and DTOs. For instance, we can have an accounting system that has accounts, ledger entries, balances and so on. The first relationship of DTO with the accounting system is that the latter is a resource for the former, so the DTO will encapsulate its conceptual model and functions in the orchestration of the business processes. Later on, however, the DTO can acquire the semantic representation of the accounting process and enact it, so the accounting system becomes a set of definitions and rules as a knowledge domain within the DTO, and therefore the original system can be deprecated. This means that there is a possibility of replacing multiple information system with DTOs. There are some scenarios where this possibility has immediate returns: when the operational and maintenance costs of the original system are high or complex, when it is in a changing domain where there is a large backlog of modification needs, and where the system configuration should be managed by end users and not by software technicians (this could be the case, for instance, of medical applications where medical doctors express meanings that are very difficult to be translated into algorithmic representations).

### 5.3 The future of DTOs

The famous science-fiction Star Trek series foresees the behavior of an organization under an advanced DTO. We can see the Enterprise Starship as a combination of four components: the Crew (that is the social system in our model), the Starship (the physical setup), the captain's deck (the Governance System) and the Logic system that governs all of the others and itself, which is the DTO. We see the Logic system in the film, because it permeates all the actions and sequences. It is controlling all aspects of the internal functioning of the ship and maintains close relationships with each crew member and specially with the captain's deck. It also governs the interactions of the crew with the ship, using wearables, sensors, pads, screens and other physical or logical devices and, more importantly, of the combination of the crew and the ship with the outer space.

In the film there is no 'Information Technologies' department. It is assumed that all engineers in the ship interact with the DTO in a dynamic adaptation of the logical system and the ship activities. We can see, using this metaphor, that we have to search for a fluid relationship between the DTO and the FS-GS system, in which the DTO is helping to accomplish it local or global missions whatever they are. DTO does not replace FS-GS nor commands it. The DTO is the universal digital framework that produces a new organizational paradigm, efficient, agile and responsive.

## 6 Conclusions and further research

In our research at UGROUND GLOBAL we have worked not only in the theoretical aspects of DTOs, but also in the actual representation of it in technical terms, building the semantic kernels, the interface mechanisms, the knowledge editors, and so on [4]. Currently our organization has used DTO technology to approach small and large digital transformations both for SME and for some large multinationals. The results are, so far, beyond our initial expectations. A DTO is a simpler, faster, more reliable way of digitalizing enterprises, and produce more efficiency, agility and performance. This is because, as was explained before, DTOs are a universal technology, and has a large applicability in most practical cases.

Now we are exploring mainly three areas of research. The first one is linked to the self-organization and autonomy requirements. We are creating the semantic representations and computational infrastructures to improve the capability of a DTO to analyze itself and adapt its performance to changing situations.

The second one is in the domain of knowledge representations. The size of the problems that we approach is large, because the objective is that all the knowledge that an enterprise has about itself can be represented into a DTO, and this can be a lot of information. Knowledge is represented in terms of propositions, templates, drawings, blueprints and recipes, that are stored in the knowledge base and transformed into interpretable definitions for the enacting engine. This research addresses the need of having highly flexible tools to store semantic definitions in many different knowledge representations.

The third one is related with the methodological framework for creating digitalized organizations. It is clear that the problems we face are new in the state-of-the-art of the organizations; society has no deep experience in digital transformation, dynamic systems and semantic-driven approaches. Our ability to deploy a universal framework for digital transformation creates methodological problems and management views that are original and innovative, and for which we have to invest energy and resources. This is not only something that applies to specific enterprises, but also to clusters of knowledge (such as functional domains or business sectors) or geographical defined problems, such as digitalization of territories, where all actors in a place are going to play new roles.

**Acknowledgements.** This work has been partially funded by the European Union's Horizon 2020 research and innovation programme under the Marie Skłodowska-Curie grant agreement 813884 (Lowcomote), by the Spanish Ministry of Science (project MASSIVE, RTI2018-095255-B-I00: CDTI project IDI-20180998), and the R&D programme of Madrid (project FORTE, P2018/TCS-4314).

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
