# OpenReview forum: "Semantic Digital Twins for Organizational Development"
_eswc-conferences.org/ESWC/2021/Workshop/SeDiT — SeDiT 2021 Oral_

### Official Review · AnonReviewer1 · 2021-03-31
**Digital Twin of an Organisation.**

**Rating:** 7
**Confidence:** 2

**Review:**

The paper discusses the concept of DTO and how they might be realised. It also proposes the use of Model Driven Engineering based on semantic technology as a way to move from the DTO to real systems and processes in an organisation.

The concepts are quite theoretical as it is early days for them so there is not much here about their use in practice beyond discussion of what could be possible.

The paper mentions continuously varying behaviour and unlimited degrees of freedom which create the impression that the approach would face severe difficulties on application given that model based approaches applied to large scale systems particularly where people are involved have proved to be problematic . It would be useful to describe early attempts at adoption even if they were limited in scope and scale.

An interesting discussion paper.

---

### Official Review · AnonReviewer5 · 2021-04-07
**Semantic Digital Twins for Organizational Development**

**Rating:** 7
**Confidence:** 4

**Review:**

The paper describes a high-level framework for the realization of an Organizational Digital Twin. The main justification for the use of a semantic digital twin approach is that a company is a dynamic system, which varies continuously due to internal and external factors, and also because of the several degrees of freedom and knowledge domains that could exist in an enterprise that is hard to be covered by existing IT tools.

The paper is interesting from the theoretical point of view, first characterizing an organization that lays the ground for the necessity for a DT, proposing some features that the DTO framework should have, and also some of the core technologies that should be used for their implementation (e.g. ontologies).

The weak point of the paper is the lack of a more detailed technical description of the implementation of this framework by means of a use case and the problems found during the realization of it.

---

### Official Review · AnonReviewer4 · 2021-04-08
**Nice theoretical argumentation  for DTOs**

**Rating:** 7
**Confidence:** 4

**Review:**

The paper summarized authors experience in the creation of DT for organization based on semantic modeling and model-driven engineering.  The paper introduces the notion of DTOs including a elaborated theoretical framework for their definition.

Even though the paper stays at a general level, it represents a very good starting point for raising discussion about DT applied to organizations. It would have been very nice having specific examples but one could understand it as future steps.

I would particularly like to see the relation with specific ontology formalisms or implementation, as well as the relation of TDOs with Web of Things models. But in general, the paper is well motivated and argued and my recommendation would be having it presented during the workshop to raise the discussion about more specific topics.

---

### Decision · Program_Chairs · 2021-04-08

Accept (Oral)